# Metabolomics to Diagnose Oxidative Stress in Perinatal Asphyxia: Towards a Non-Invasive Approach

**DOI:** 10.3390/antiox10111753

**Published:** 2021-11-02

**Authors:** Anne Lee Solevåg, Svetlana N. Zykova, Per Medbøe Thorsby, Georg M. Schmölzer

**Affiliations:** 1The Department of Paediatric and Adolescent Medicine, Oslo University Hospital, 0424 Nydalen, Norway; 2Biochemical Endocrinology and Metabolism Research Group, The Hormone Laboratory, Department of Medical Biochemistry, Oslo University Hospital, 0424 Nydalen, Norway; svezyk@ous-hf.no (S.N.Z.); pertho@ous-hf.no (P.M.T.); 3Centre for the Studies of Asphyxia and Resuscitation, Neonatal Research Unit, Royal Alexandra Hospital, Edmonton, AB 23821, Canada; georg.schmoelzer@me.com; 4Department of Pediatrics, Faculty of Medicine and Dentistry, University of Alberta, Edmonton, AB 23821, Canada

**Keywords:** asphyxia neonatorum, non-invasive diagnostics, metabolomics, oxidative stress, saliva

## Abstract

There is a need for feasible and non-invasive diagnostics in perinatal asphyxia. Metabolomics is the study of small molecular weight products of cellular metabolism that may, directly and indirectly, reflect the level of oxidative stress. Saliva analysis is a novel approach that has a yet unexplored potential in metabolomics in perinatal asphyxia. The aim of this review was to give an overview of metabolomics studies of oxidative stress in perinatal asphyxia, particularly searching for studies analyzing non-invasively collected biofluids including saliva. We searched the databases PubMed/Medline and included 11 original human and 4 animal studies. In perinatal asphyxia, whole blood, plasma, and urine are the most frequently used biofluids used for metabolomics analyses. Although changes in oxidative stress-related salivary metabolites have been reported in adults, the utility of this approach in perinatal asphyxia has not yet been explored. Human and animal studies indicate that, in addition to antioxidant enzymes, succinate and hypoxanthine, as well acylcarnitines may have discriminatory diagnostic and prognostic properties in perinatal asphyxia. Researchers may utilize the accumulating evidence of discriminatory metabolic patterns in perinatal asphyxia to develop bedside methods to measure oxidative stress metabolites in perinatal asphyxia. Although only supported by indirect evidence, saliva might be a candidate biofluid for such point-of-care diagnostics.

## 1. Introduction

Failed placental gas exchange or deficient gas exchange in the lungs after birth may cause perinatal asphyxia with hypoxia and hypercapnia resulting in mixed metabolic and respiratory acidosis. Asphyxiated infants can present with severe cardiorespiratory compromise at birth and a need for cardiopulmonary resuscitation with supplementary oxygen. However, mild symptoms of asphyxia may also result in later morbidity and mortality in affected infants [1,2].

In perinatal asphyxia, balancing the harmful effects of iatrogenic hyperoxia (‘oxidative stress’) vs. anaerobic metabolism (continuing/prolonged hypoxia) is a complex task. Although humans have physiological and biochemical defense mechanisms to prevent hypoxia [3], defense mechanisms against hyperoxia are less developed in newborn infants, with the resulting oxidative stress potentially causing organ injury.

During asphyxia and resuscitation, disrupted cellular homeostasis causes significant metabolic changes [4], and studies of the metabolome may provide a pathophysiological ‘snapshot’ of the condition. Metabolomics is the study of small molecular weight (<1500 Da) endogenous metabolites present in tissues or biofluids typically at concentrations above 1 uM, and may be used to directly and indirectly measure oxidative stress [5]. Measuring oxidative stress in the neonate could aid decisions to initiate time-critical interventions including therapeutic hypothermia but poses challenges due to difficult sampling as well as a need for quick and repeated analyses [6].

### Saliva as a Promising Simple and Non-Invasively Collected Biofluid

In the search for therapeutic target molecules, saliva has been used as a matrix in genomics [7], proteomics [8], and metabolomics [9]. Non-invasive and safe real-time sampling, and ease of collection, handling, storage, and post-storage contribute to its attractiveness and relatively low costs. However, different collection methods might yield different analysis results. Thus, it is important to standardize the collection, e.g., either by passive drainage or by specific collection devices [8]. In newborn infants, saliva may have particular advantages over invasively collected biofluids including blood. Compared to blood sampling, saliva sampling is not painful and does not deplete patients with blood cells and nutrients.

The aim of this review paper is to provide an overview of metabolomics studies of oxidative stress in perinatal asphyxia, with particular emphasis on analysis methods and biofluids used. Measuring the level of oxidative stress might aid in diagnosing infants with significant perinatal asphyxia but a mild initial clinical presentation. We were particularly interested in evidence about feasible and non-invasive diagnostics, with saliva being a novel biofluid with yet unknown potential in perinatal asphyxia.

## 2. Materials and Methods

### Search Strategy and Selection Criteria

We did a non-systematic search in PubMed/Medline with no limits on publication date. The search was finished in September 2021 and included the terms “metabolomics” AND “oxidative stress” AND “neonatal asphyxia” OR “perinatal asphyxia” OR “HIE” OR “hypoxic ischemic encephalopathy”. Reference lists of relevant articles were hand searched for additional publications of interest. Only English-language, peer-reviewed studies were included. We excluded studies that analyzed oxidative stress in tissues, metabolomics studies with a lack of focus on oxidative stress, as well as review articles, case reports, and conference abstracts. We included studies in term and preterm infants, as well as animal studies. The search identified 11 original human studies and 4 original animal studies.

## 3. Results

### 3.1. Methods for Metabolomics Analyses

Metabolomics analyses can be targeted and untargeted [10], the latter investigating both known and unknown metabolites. Analytical methods for determination of small molecular weight metabolites in biofluids include (i) immunological methods including radioimmunoassay and enzyme-linked immunosorbent assay (targeted), (ii) high-resolution ^1^H (proton) and ^13^C nuclear magnetic resonance (NMR) spectroscopy [11] (targeted and untargeted), (iii) gas chromatography (GC) coupled to mass spectrometry (MS) (targeted and untargeted), (iv) (ultra-performance) liquid chromatography coupled to MS (LC-MS) (targeted and untargeted), (v) infrared and Raman spectroscopy [12,13], and (vi) capillary electrophoresis–time-of-flight mass spectrometry (CE–TOFMS) (semi-targeted).

MS-based assays are considered more accurate than immunological methods (Table 1), with LC-MS having advantages over GC-MS as it does not require as long derivatization processes that may cause measurement error. Moreover, ultra-high performance (UP) liquid chromatography tandem mass spectrometry (UPLC-MS/MS) enables rapid chromatographic separations. UPLC-MS/MS is characterized by good selectivity and sensitivity, and a high sample throughput [6]. The metabolomics MS-workflow for biofluids including saliva is presented in Figure 1.

### 3.2. Different Biofluids

#### 3.2.1. Blood

Blood is the biological material for most diagnostic tests in clinical routine. Blood reflects the dynamic, real-time metabolic response to asphyxia and reoxygenation, and allows for quick analysis. However, blood sampling in neonates may be technically challenging, is invasive, and painful. Blood sampling may contribute to anemia in more premature infants as their blood volume is lower. Indeed, Sachse et al. [11] were able to perform ^1^H NMR metabolomics analysis with only 250 μL of piglet plasma, and Sanchez-Illana et al. [6] used 100 μL plasma for UPLC-MS/MS analysis. Dekker et al. [15] used 300 μL full blood from premature infants.

An alternative blood sample option is umbilical cord blood, which is non-invasive and painless and may allow the collection of larger volumes to perform several analyses. However, umbilical cord blood only provides information until the time of birth. Thus, combined measurements of metabolites in umbilical cord blood and subsequently in the infant’s blood are needed.

There is increasing evidence that preanalytical factors may heavily influence the results in metabolomics, both in the individual investigated and handling of the sample before metabolomic analyses. Currently, the need for access to advanced sample preparation is needed when conducting metabolomic studies [16]. Dried blood spot samples have been suggested for metabolomics analysis as they do not require preanalytical preparation, special storage, or freezing [17]. Several studies indicate the utility of blood spots for metabolomics analysis, with dried blood spots being largely equivalent to protein-precipitated plasma [18,19].

#### 3.2.2. Urine

Although urine collection might be an easy alternative, the urine output in the first days after birth is typically low, and infants with perinatal asphyxia often have impaired renal function with oliguria or anuria. Urine for metabolomics analysis must be collected under sterile conditions because bacterial metabolism influences the urine metabolome, and samples must be frozen at −80 °C immediately after collection [20]. Urine is suitable for repeat analysis and assessment of markers of lipid peroxidation up to days or weeks after an insult or intervention, with the associated disadvantage of limited time resolution. Thus, urine analysis might not be the preferred option when the purpose is to institute time-critical interventions.

Sachse et al. [11] used 550 μL of piglet urine for ^1^H NMR metabolomics analysis. They demonstrated a different time profile of plasma versus urine metabolites and concluded that renal handling of different metabolites may vary and needs to be considered if urine is used instead of plasma in metabolomic analyses.

#### 3.2.3. Cerebrospinal Fluid

Cerebrospinal fluid (CSF) is used in the diagnosis and management of neurological diseases. CSF is produced by plasma ultrafiltration and membrane secretion, is nearly free from cells, and protein levels are usually very low. However, CSF must be collected through a lumbar puncture, which is invasive and may be technically difficult in neonates. The quantity of fluid obtained may be limited and hemorrhage caused by the puncture may preclude interpretation. Thus, CSF metabolomics analyses have been performed in the experimental [21], but only rarely in the clinical setting of perinatal asphyxia [22].

#### 3.2.4. Saliva

Although saliva can be collected non-invasively, the quantity may be limited. However, technological advances enable the analysis of small sample volumes (Table 2). Yen et al. [23] state that suctioning of the mouth typically yields 10–50 μL neonatal saliva, whereas sponges and wick applicators may yield slightly higher volumes, directly correlated to collection time. Oxidative stress markers glutathione [24], malondialdehyde [24,25,26,27] catalase, protein carbonyls, glutathione peroxidase, and 8-hydroxy-2′-deoxyguanosine [27], as well as isoprotanes, isofuranes, neuroprostanes, and neurofuranes [28] have all been measured in saliva from adults. However, although sporadically mentioned as an alternative [5,29], the evidence of metabolomics analysis of saliva in asphyxiated neonates is sparse, if not non-existing.

A selection of metabolites related to oxidative stress is presented in Table 2.

### 3.3. Animal Studies

In animal models of perinatal asphyxia, 100% oxygen was associated with metabolic markers of delayed cellular recovery and increased oxidative stress [30,31,32]:

In a piglet model of perinatal asphyxia, baseline urine metabolome, analyzed with ^1^H NMR spectroscopy, distinguished between piglets that later became asystolic or not. The post-resuscitation-, seen in relation to the baseline metabolome, differentiated between piglets reoxygenated with different oxygen concentrations, primarily due to variations in metabolites with free radical scavenging properties: urea, creatinine, and malonate, as well as methylguanidine and hydroxyisobutyric acid [31]. A few years later, the same group investigated the same oxygen concentrations and confirmed distinct urine metabolomic patterns in piglets reoxygenated with 18, 21, 40, and 100% oxygen, respectively. Alanine and succinate were elevated, but glycine was unchanged in asphyxiated piglets resuscitated with 21% oxygen [32].

Moreover, in asphyxiated piglets and using flow injection analysis, MS/MS and LC-MS/MS, Solberg et al. [30] measured reduced blood succinate, fumarate, and alpha keto-glutarate indicating an earlier mitochondrial recovery when 21% oxygen was used for reoxygenation compared to 100% oxygen. Furthermore, oxysterols and acylcarnitine showed a differential response to 21% versus 100% oxygen reoxygenation. The ratios of alanine to branched chained amino acids, and of glycine to branched chained amino acids correlated with the duration of hypoxia.

Sachse et al. [11] reported a strong and consistent increase in alanine, succinate, hypoxanthine, and branched-chain amino acids in 125 asphyxiated piglets. There was a decrease of ^1^H NMR signals associated with lipids. Baseline plasma hypoxanthine and lipoprotein concentrations were inversely correlated to the duration of hypoxia sustained before asystole occurred, but there was no evidence for a differential metabolic response to different resuscitation protocols including the use of supplementary oxygen, or in terms of survival [11]. In urine, branched-chain amino acids, especially valine, but also alanine and choline concentrations were higher after asphyxia and resuscitation compared to baseline.

### 3.4. Clinical Studies

Studies have comprehensively examined the metabolome in hypoxic-ischemic encephalopathy (HIE), but with a minor focus on oxygenation and oxidative stress [33,34]. A study of initial ventilation with 30% vs. 90% oxygen provided insights into oxidative stress markers in blood and urine of preterm infants ≤28 weeks of gestation, but not related to perinatal asphyxia [35].

#### 3.4.1. Response to Hyperoxia

Vento et al. [36,37,38,39], showed that the blood reduced (GSH)-to-oxidized glutathione (GSSG) ratio was lower in asphyxiated infants that received 100% oxygen compared to 21% oxygen, lasting up to four weeks after birth [37]. Hundred percent oxygen ventilation was associated with increased activity of antioxidant enzymes including superoxide dismutase and glutathione redox cycle enzymes [38,39]. Urine N-acetyl-glucosaminidase correlated with GSSG and was significantly higher in infants that received 100% oxygen [36].

Dekker et al. [15] recently investigated initial ventilation with 30% versus 100% oxygen in preterm infants <30 weeks’ gestation and found no difference in the lipid peroxidation product 8-iso-prostaglandin in umbilical cord blood, or infant blood at 1 and 24 h of age between 0% versus 100% oxygen.

#### 3.4.2. Prognostic Utility

Negro et al. [40] measured blood advanced oxidation protein products (AOPP), non-protein bound iron (NPBI), and F2-isoprostanes (F2-IsoPs) in 84 infants with severe versus mild/moderate HIE at three different time points: P1 (4–6 h), P2 (24–72 h), and P3 (5 days). Mean (standard deviation) values of AOPP, NPBI, but not F2-IsoPs were significantly higher in infants with severe HIE with AOPP 34.1(39.2) vs. 15.7(15.5), *p* = 0.033 and NPBI 3.9 (4.4) versus 1.1(2.5), *p* = 0.013 at P1. However, there was no difference between groups at P2 and P3. A regression model showed that AOPP levels and male sex were both risk factors for higher brain damage scores; AOPP: OR 3.6, 95% confidence interval (1.1–12.2) and sex: OR 5.6, 95% confidence interval (1.2–25.7), respectively.

Vasiljevic et al. [22] performed routine lumbar puncture in 90 infants with HIE and reported a good relationship of glutathione peroxidase activity with the clinical stage of HIE (*p* < 0.0001) and gestational age (*p* < 0.0001). Glutathione peroxidase activity in CSF corresponded to later neurodevelopment outcome (*p* < 0.001) and showed a strong correlation with CSF levels of neuron-specific enolase (*p* < 0.001), a biomarker of the extent of brain injury.

Umbilical cord blood from term infants with confirmed HIE (*n* = 31) was compared to asphyxiated infants without encephalopathy (*n* = 40), and matched controls (*n* = 71) [41]. Targeted metabolomics revealed a significant increase in 29 metabolites from 3 distinct classes (amino acids, acylcarnitines, and glycerophospholipids) in infants with HIE or asphyxia compared to controls. Moreover, eight amino acids significantly increased in infants with HIE, but not asphyxia or matched controls. Thirteen acylcarnitines were significantly increased in infants with HIE and asphyxia without HIE, but the changes were more pronounced in the HIE population. A logistic regression model using five metabolites clearly delineated the severity of asphyxia and classified HIE infants with AUC = 0.92. These data suggest significant disruption to the energy pathways as well as nitrogen and lipid metabolism in both asphyxia and HIE.

Chu et al. [42] analyzed urine metabolite profiles from 256 asphyxiated infants and reported a positive relationship between suppressed biochemical networks involved in the macromolecular synthesis and perinatal asphyxia associated with significant oxidative stress and morbidity. In particular, elevated organic acids related to oxidative stress were significantly associated with neurodevelopmental outcomes: Ethylmalonate, 3-hydroxy-3-methylglutarate, 2-hydroxy-glutarate and 2-oxo-glutarate were associated with a good outcome; whereas glutarate, methylmalonate, 3-hydroxy-butyrate, and orotate were associated with a poor outcome. In preterm asphyxiated infants, urine threonine and 3-hydroxyisovalerate were increased; and dimethylglycine, dimethylamine, creatine, succinate, formate, urea, and aconitate were decreased [43]. These data demonstrated the potential application of bioinformatics methods in this metabolomics study and its potential clinical relevance.

Reinke et al. [44] proposed a metabolomic index ((succinate x glycerol) / (β-hydroxybutyrate × O-phosphocholine)) to identify asphyxiated infants at risk of developing HIE, as elevated β-hydroxybutyrate, glycerol, O-phosphocholine, and succinate were most strongly associated with HIE severity. Ahearne et al. [45] later showed that if this index was <0.13, the outcome was likely to be normal (sensitivity 65% and specificity of 91%), whereas an index >2.4 was associated with a “severe outcome” (sensitivity of 80% and specificity of 100%). The authors concluded that this metabolomic index might be used in identifying neonates at risk of developing severe HIE.

## 4. Discussion

With a focus on oxidative stress, human and animal studies indicate that Krebs cycle intermediates including succinate, and hypoxanthine, in addition to acylcarnitines, play a role in perinatal asphyxia and may have prognostic and discriminatory properties.

The neonatal brain is highly oxygen dependent with low antioxidant capacity. This, combined with a high content of unsaturated fatty acids, makes the neonatal brain particularly vulnerable to oxidative stress [46]. Hypoxanthine is a purine metabolite, and levels are elevated during hypoxia. Hypoxanthine is oxidized in the presence of xanthine oxidase to uric acid during reoxygenation [47]. This generates an oxygen-free radical burst that, although uric acid has antioxidant properties, overwhelms endogenous antioxidant defense systems. Excessive production of reactive oxygen species [48] including hydrogen peroxide, hydroxyl free radical, superoxide anion radical, and reactive nitrogen species [49], proteases and caspases [50] in turn cause mitochondrial dysfunction and cell death. Free radical-induced lipid (arachidonic acid) peroxidation results in the generation of prostaglandin-like (prostanoids) isoprostanes [51,52]. Isoprostanes are often considered the gold standard for in vivo measurement of lipid peroxidation [53,54,55]. However, biofluid isoprostanes have demonstrated limited discriminatory properties in asphyxiated infants.

In asphyxia, incomplete fatty acid oxidation may result in an increase of fatty acid coenzyme A esters which bind to carnitine, resulting in the production of acylcarnitines [56]. High carnitine suppresses lipid peroxidation and the production of hydroxyl free radicals [57]. Carnitine thus has potent antioxidant activity [58], whereas acylcarnitines in perinatal asphyxia may indirectly reflect oxidative stress. Superoxide dismutase, glutathione, catalase, and glutathione peroxidase are cellular antioxidants [49]. Debuf et al. [29] performed a review of biomarkers of perinatal asphyxia and highlighted acylcarnitines as the most promising.

Several reviews, including one very recent [29], have addressed metabolomics biomarker analysis in perinatal asphyxia and HIE. However, they did not focus on oxidative stress the way our present review does. Perinatal asphyxia is associated with increased metabolites derived from lipid peroxidation. However, lipid peroxidation products seem to have limited utility in identifying infants at risk of developing HIE. Therefore, it has been proposed that the most reliable way to use metabolomics in perinatal asphyxia risk stratification and prognostication, is the establishment of a metabolic index composed of multiple metabolites that together have better prognostic value.

Biomarkers used for screening should be easy to collect, have a fast laboratory turnaround time, be reliable, and relatively inexpensive. Modern methods, e.g., LC-MS/MS have a turnaround time ranging from minutes to hours, opening for the possibility of integration into care. As stated by Debuf et al. [29], despite a current lack of point-of-care methods to measure these metabolites, considering the increasing advances in metabolomics, a “bench to the bedside” approach may be realistic within a reasonable time frame. Mussap et al. [5] suggested the translation of research results into the development of a low-cost device, e.g., a dipstick.

UPLC-MS/MS has been used to measure oxidative stress markers in saliva from adult patients and, although not supported by the available literature, saliva might be a suitable biofluid for point-of-care diagnostics. Saliva is, in theory, easy to collect and collection is non-invasive. However, saliva collection in newborns could be challenging, partially due to the limited amounts acquired. Standardized approaches to saliva collection, processing, and analyses are required for proper interpretation.

Strengths of this review include the innovative way of using existing knowledge to provide directions for research to develop non-invasive rapid tests of oxidative stress in perinatal asphyxia. Limitations include that we identified and report markers of oxidative stress in a wider sense and that we use indirect evidence, i.e., animal and adult data, to support our conclusions. However, we believe that the transparency of the reporting of results also leaves room for the readers to conclude from the limited knowledge base themselves.

In conclusion, in perinatal asphyxia, evidence about discriminatory metabolites related to oxidative stress, single or in combination, could be used to develop methods for rapid diagnostic and risk assessment purposes. Saliva might be a candidate biofluid for such point-of-care methods, however, more research is needed.

## Figures and Tables

**Figure 1 antioxidants-10-01753-f001:**
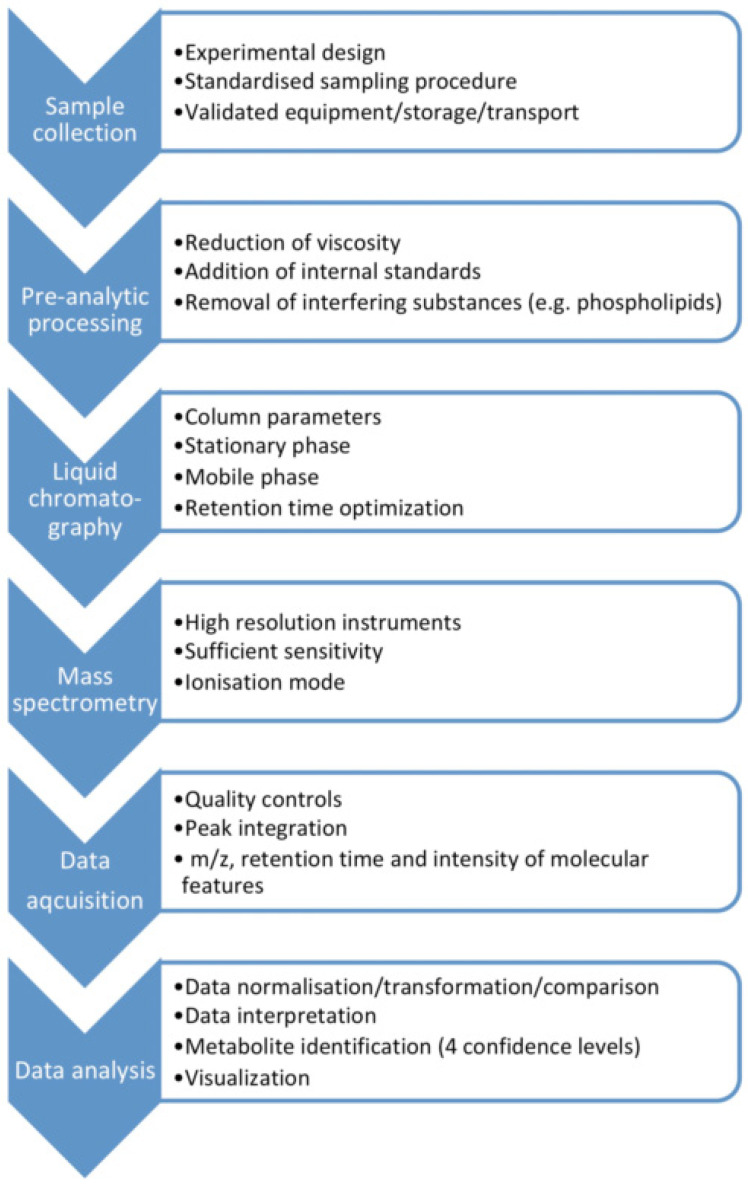
The metabolomics mass spectrometry workflow for biofluids including saliva.

**Table 1 antioxidants-10-01753-t001:** Advantages and disadvantages of mass spectrometry-based assays and immunoassays [14].

Method	Advantages	Disadvantages
Mass spectrometry	High selectivity	Limited to some laboratories
	High sensitivity	Limited sample throughput
	High throughput	Limited user friendliness, requires specialized personnel
	Requires low sample volumes	High equipment costs
	Multiplexing possible	Sample complexity issues
	Relatively low running costs	
	Not restricted to biomolecules	
	High intra- and inter-assay reproducibility	
Immunoassay	Low training requirements	Limited selectivity
	Kits available from commercial vendors	Limited analyte detection abilities
	Validated and approved	Requires relatively high sample volumes
	Relatively low-cost equipment	Relatively expensive reagents
	Relatively high throughput	Relatively high intra- and interassay and laboratory variability
	Relatively high sensitivity	Relatively long assay time

**Table 2 antioxidants-10-01753-t002:** Metabolites related to oxidative stress [29,30,31,32]. Reproduced with permission from Dr. Wishart, University of Alberta, Edmonton, Canada.

Metabolite	Description
Urea	The principal end product of protein catabolism.
Creatinine	A breakdown product of creatine phosphate in muscle.
Malonic acid	Malonic acid, also known as malonate or H2MALO, belongs to the class of organic compounds known as dicarboxylic acids and derivatives. In humans, malonic acid is involved in fatty acid biosynthesis
Methylguanidine	A guanidine compound deriving from protein catabolism. Synthesized from creatinine concomitant with the synthesis of hydrogen peroxide from endogenous substrates in peroxisomes. A nitric oxide synthase inhibitor.
L-Alanine	Alanine (Ala), also known as L-alanine, is an alpha-amino acid. Glutamate can transfer its amino group to pyruvate, a product of muscle glycolysis, through the action of alanine aminotransferase, forming alanine and alpha-ketoglutarate. Plasma alanine is often decreased when branched chain amino acids (BCAA) are deficient.
Succinate	Succinic acid (succinate) is a dicarboxylic acid.
Fumaric acid	Fumaric acid is a dicarboxylic acid.
Alpha-ketoglutarate	Oxoglutaric acid, also known as alpha-ketoglutarate, alpha-ketoglutaric acid, AKG, or 2-oxoglutaric acid. AKG is a nitrogen scavenger.
Hydroxycholesterol	27-Hydroxycholesterol (27-HC), also known as (25R)-cholest-5-ene-3β,26-diol or by its conventional name 26-hydroxycholesterol.
S-Adenosylmethionine	S-Adenosylmethionine, also known as SAM or acylcarnitine, belongs to the class of organic compounds known as 5’-deoxy-5’-thionucleosides. Possesses anti-inflammatory activity.
Glycine	An alpha-amino acid. Glycine is involved in the body’s production of DNA, hemoglobin, and collagen, and in the release of energy.
Hypoxanthine	Hypoxanthine, also known as purine-6-ol or Hyp, belongs to the class of organic compounds known as purines. Under normal circumstances hypoxanthine is readily converted to uric acid.
Valine	Valine (Val) or L-valine is an alpha-amino acid. L-valine is a BCAA. The BCAAs consist of leucine, valine, and isoleucine (and occasionally threonine).
Choline	Important as a precursor of acetylcholine, as a methyl donor in various metabolic processes, and in lipid metabolism.
Glutathione	Like cysteine, glutathione contains the crucial thiol (-SH) group that makes it an effective antioxidant.
Ethylmalonate	Ethylmalonic acid, also known as alpha-carboxybutyric acid or ethylmalonate, is a branched fatty acid. Ethylmalonic acid can be synthesized from malonic acid.
3-Hydroxymethylglutaric acid	3-Hydroxymethylglutaric acid is an “off-product” intermediate in the leucine degradation process.
Glutaric acid	Is produced during the metabolism of some amino acids, including lysine and tryptophan.
Methylmalonic acid	Methylmalonic acid is a malonic acid derivative, which is a vital intermediate in the metabolism of fat and protein.
Threonine	Threonine (Thr) or L-threonine is an alpha-amino acid. Threonine is sometimes considered a BCAA. Threonine is metabolized in at least two ways. In many animals it is converted to pyruvate via threonine dehydrogenase. An intermediate in this pathway can undergo thiolysis with CoA to produce acetyl-CoA and glycine.
3-Hydroxyisovaleric acid	A byproduct of the leucine degradation pathway.
Dimethylglycine	Dimethylglycine (DMG) is an amino acid derivative. The human body produces DMG when metabolizing choline into glycine. Homocysteine and betaine are converted to methionine and N,N-dimethylglycine by betaine-homocysteine methyltransferase.
Dimethylamine	An organic secondary amine.
Creatine	A naturally occurring non-protein compound classified as ‘alpha amino acids and derivatives’. Its primary metabolic role is to combine with a phosphoryl group, via the enzyme creatine kinase, to generate phosphocreatine, which is used to regenerate ATP. It is naturally produced in the human body from the amino acids glycine and arginine, with an additional requirement for methionine to catalyze the transformation of guanidinoacetate to creatine.
Formic acid	The simplest carboxylic acid. Inhibition of cytochrome oxidase by formate may cause cell death by increased production of cytotoxic reactive oxygen species secondary to the blockade of the electron transport chain).

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
