# Peer review of "Metabolomics to Diagnose Oxidative Stress in Perinatal Asphyxia: Towards a Non-Invasive Approach"

_antioxidants, 2021, doi:10.3390/antiox10111753_

Round 1

Reviewer 1 Report

This overview deals with advances in omics of saliva metabolites for diagnosis of OS during perinatal asphyxia.

REMARKS

  1. In introduction, I miss to add some info about saliva sample to be a bodily fluid taken noninvasively and suitable to search for therapeutic target molecules. This search is based on use of omics approaches such as proteomics or metabolomics. Please, follow the lately published review papers to be added representing proteomics and metabolomics of saliva.

https://doi.org/10.1515/biol-2018-0023

https://doi.org/10.1016/j.ajoms.2021.02.003

  1. Please, add the whole range of search in the databases.

  1. MS-based methods vs. IAs pose several pros-and-cons when used in omics workflow, please list them.

  1. Draw a metabolomics workflow for saliva analysis and provide it as a picture.

  1. In conclusion, what are your plans in this scope of investigation, in the future?

Author Response

  1. In introduction, I miss to add some info about saliva sample to be a bodily fluid taken noninvasively and suitable to search for therapeutic target molecules. This search is based on use of omics approaches such as proteomics or metabolomics. Please, follow the lately published review papers to be added representing proteomics and metabolomics of saliva.

https://doi.org/10.1515/biol-2018-0023

https://doi.org/10.1016/j.ajoms.2021.02.003

thank you - this has been added

  1. Please, add the whole range of search in the databases.

thank you - this has been added

  1. MS-based methods vs. IAs pose several pros-and-cons when used in omics workflow, please list them.

thank you - this has been added

  1. Draw a metabolomics workflow for saliva analysis and provide it as a picture.

thank you - this has been added

  1. In conclusion, what are your plans in this scope of investigation, in the future?

thank you - We believe that during/after perinatal asphyxia, saliva could be used to assess oxidative stress as rapid diagnostic and risk assessment tool. 

Reviewer 2 Report

The title and introduction of the review focus on the use of saliva as an alternate biofluid for diagnosing oxidative stress in perinatal asphyxia. However, as noted by the authors, metabolomics studies of saliva are lacking. Reading the review, it would be more accurate to say that it is a review of the current state of metabolomics studies to evaluate perinatal asphyxia or HIE in blood or urine. I think that the title and introduction need to be changed to reflect what is reported in the review and the potential use of saliva can be introduced in the discussion. A few specific comments related to the above are provided below.

  1. Line 31 – Failed placental gas exchange would obviously take place prior to birth; have studies evaluated oxidate stress in the maternal saliva metabolome?
  2. Section 3.2.4 - How much saliva is produced by the neonate? Other sections noted how much blood or urine was used for the referenced analyses. How much saliva was used in the reported studies?
  3. In Table 1, why are Malonic Acid, L-Alanine, and Formic Acid in bold?
  4. Section 3.3 – All of the animal studies reported evaluated either urine or blood. These are not applicable to the review since none evaluated metabolites in the saliva.
  5. Section 3.4 – The referenced clinical studies also did not evaluate the saliva metabolome.
  6. Line 265-266 – The authors write, “they have not taken a strictly “oxidative stress” focus like our present review”. This does seem to be the focus of the review rather than the use of saliva to evaluate oxidative stress. The authors should consider a new title and updated introduction to reflect this focus. They could still propose in the discussion the use of saliva as an alternate biofluid but as they have noted and is clear in the review, there is a lack of data regarding asphyxia and the oxidative stress markers in the saliva.

Author Response

  • Line 31 – Failed placental gas exchange would obviously take place prior to birth; have studies evaluated oxidate stress in the maternal saliva metabolome?
    thank you - this has been added/edited

  • Section 3.2.4 - How much saliva is produced by the neonate? Other sections noted how much blood or urine was used for the referenced analyses. How much saliva was used in the reported studies?

thank you - this has been added/edited

  • In Table 1, why are Malonic Acid, L-Alanine, and Formic Acid in bold?

thank you - this has been edited

  • Section 3.3 – All of the animal studies reported evaluated either urine or blood. These are not applicable to the review since none evaluated metabolites in the saliva.

thank you - this has been edited

  • Section 3.4 – The referenced clinical studies also did not evaluate the saliva metabolome.

thank you - this has been edited

  • Line 265-266 – The authors write, “they have not taken a strictly “oxidative stress” focus like our present review”. This does seem to be the focus of the review rather than the use of saliva to evaluate oxidative stress. The authors should consider a new title and updated introduction to reflect this focus. They could still propose in the discussion the use of saliva as an alternate biofluid but as they have noted and is clear in the review, there is a lack of data regarding asphyxia and the oxidative stress markers in the saliva.

thank you - this has been edited

Round 2

Reviewer 1 Report

Authors have reacted to all queries given.

Reviewer 2 Report

This is a nice summary of the current use of metabolomics to diagnose asphyxia and how it might be applied to diagnose perinatal asphyxia. The authors also clearly show how saliva could be an appropriate biofluid to evaluate oxidative stress in newborn infants and thus drive point-of-care diagnostics.